# Alternative Splicing Analysis Revealed the Role of Alpha-Linolenic Acid and Carotenoids in Fruit Development of *Osmanthus fragrans*

**DOI:** 10.3390/ijms24108666

**Published:** 2023-05-12

**Authors:** Cancan Ma, Cheng Zhang, Xiaoyan Wang, Fuyuan Zhu, Xianrong Wang, Min Zhang, Yifan Duan

**Affiliations:** Co-Innovation Center for Sustainable Forestry in Southern China, College of Biology and the Environment, International Cultivar Registration Center for Osmanthus, Nanjing Forestry University, Nanjing 210037, China

**Keywords:** *Osmanthus fragrans*, alternative splicing, fruit development, alpha-linolenic acid metabolism, carotenoid biosynthesis

## Abstract

Alternative splicing refers to the process of producing different splicing isoforms from the same pre-mRNA through different alternative splicing events, which almost participates in all stages of plant growth and development. In order to understand its role in the fruit development of *Osmanthus fragrans*, transcriptome sequencing and alternative splicing analysis was carried out on three stages of *O. fragrans* fruit (*O. fragrans* “Zi Yingui”). The results showed that the proportion of skipping exon events was the highest in all three periods, followed by a retained intron, and the proportion of mutually exclusive exon events was the lowest and most of the alternative splicing events occurred in the first two periods. The results of enrichment analysis of differentially expressed genes and differentially expressed isoforms showed that alpha-Linolenic acid metabolism, flavonoid biosynthesis, carotenoid biosynthesis, photosynthesis, and photosynthetic-antenna protein pathways were significantly enriched, which may play an important role in the fruit development of *O. fragrans*. The results of this study lay the foundation for further study of the development and maturation of *O. fragrans* fruit and further ideas for controlling fruit color and improving fruit quality and appearance.

## 1. Introduction

Alternative splicing (AS) refers to the process in which pre-mRNA finally produces a variety of different transcription products by changing splicing sites during gene expression, which is an important regulatory mechanism during gene expression [1]. Alternative splicing was first described in the study of the adenovirus hexon gene, which showed that a single gene could produce multiple mRNAs with different functions [2]. The discovery overturned the theory of molecular biology at the time that “one gene corresponds to one protein.” Since then, alternative splicing, as a post-transcriptional processing mechanism of genes, has been found to exist widely in eukaryotes and is the main form of regulating gene expression in eukaryotes [3]. Alternative splicing of pre-mRNA was believed to be one of the important causes of protein functional diversity, which allows a gene to encode multiple different transcriptional and protein products, greatly increasing protein diversity and the complexity of gene expression, and was an important factor in regulating gene expression and generating proteomic diversity [4].

There are five main types of alternative splicing events, including skipped exon, mutually exclusive exon, alternative 5′ splice site, alternative 3′ splice site, and retained intron [5]. The proportion of variable splicing events in genes varies among species, and species in the same community have the same types of major variable splicing events. The main type of alternative splicing events in fungi and protists is intron retention. The frequency of skipped exon events was also higher in plants than in fungi and protozoa [6]. Many results indicated that alternative splicing played an important role in plant growth and development [7,8,9]. The potential regulatory network created by alternative splicing under anoxic conditions during *Oryza sativa* germination could facilitate rice agricultural production suitable for direct seeding systems, and AS induced by moderate soil desiccation provides a possible mechanism for the regulation of microRNA processing, which may also contribute to increased grain filling in the lower spikelets [10,11]. Many genes involved in substance metabolism have been demonstrated to undergo alternative splicing at different developmental stages and under stress conditions. Alternative splicing, as a regulatory mechanism, fine-tunes plant metabolism by altering the biochemical activities, interactions, and subcellular localization of proteins encoded by spliced isoforms of various genes [12]. Abscisic acid treatment has a broad impact on AS patterns, manifested as an increase in the number of unconventional splice sites and increased protein diversity [13]. With the development of high-throughput sequencing technology and the improvement of sequencing depth, more and more alternative splicing events in fruit development were detected. The number of skipped exon events increased significantly during this period, which indicated that exon-skipping events may be involved in the regulation of fruit ripening and senescence in *Actinidia chinensis* [14]. The CsNES gene of *Camellia sinensis* underwent alternative splicing, and its products had catalytic activity and can regulate terpene metabolism [15]. Alternative splicing was ubiquitous in the whole development of the fruit of ornamental crabapple, and these spliced genes in the development process of ornamental crabapple are mainly related to phosphorylation and sugar metabolism [16].

In this study, transcription data on three different stages of fruit development of *O. fragrans* were used to analyze the types and quantities of alternative splicing; GO and KEGG analyses were performed on the differentially spliced genes and differentially spliced isoforms, hoping to unravel the role of AS during fruit development or color shift and provide the basis for further research of *O. fragrans*.

## 2. Results

### 2.1. Phenotype and AS Event Statistics

According to fruit color of *O. fragrans*, samples were collected at Nanjing Forestry University in different time periods. The samples were divided into three stages: during the development of *O. fragrans* fruit (*O. fragrans* “Zi Yingui”), from the first stage to the third stage, color of *O. fragrans* fruit changed from green, half green, and half purple to full purple, and the color deepened continuously and fruit evolved from being firm to tender (Figure 1A). In this project, a total of nine samples were tested using public DNBSEQ platform, and the samples at each stage had three biological replicates, with an average output of 6.41 Gb of data per sample. A total of 417,176,396 sequencing tags were obtained, and an average of 42,706,801 tags were obtained for each sample. After filtering, the tags with sequencing quality higher than Q20 of each sample were greater than 96% (Appendix A). The average matching rate of samples against genomes was 87.08% and the average matching rate against gene sets was 69.48%. There were 3533 novel genes predicted; 39,674 expressed genes were detected, including 36,224 known genes and 3450 predicted novel genes. A total of 26,595 novel transcripts were detected, including 23,062 novel alternative splicing isoforms of known protein-encoding genes and 3533 novel protein-encoding genes (Appendix A).

Five alternative splicing events were detected, including skipped exon (SE), the alternative 5′ splicing site (A5SS), the alternative 3′ splicing site (A3SS), mutually exclusive exon (MXE), and retained intron (RI) (Figure 1B). In this study, we analyzed the alternative splicing events of *O. fragrans* fruit transcription genes and identified 85,781 alternative splicing events in *O. fragrans* fruit. According to the types of alternative splicing events, the number of SE was the highest, 15,516, accounting for 52.43% of all the alternative splicing events, and 13,074, accounting for 15.24% of all the alternative splicing events at A3SS. A total of 7936 (9.25%) sites were variable at A5SS. There were 2524 MXEs in total, accounting for 2.94%. The number of RI was 17,274 (20.14%) (Figure 1C, Appendix A). Most of the alternative splicing events took place in the first two stages (Figure 1D).

### 2.2. Analysis of Differentially Expressed Alternative Splicing Events

Differentially expressed genes (DEGs) and differentially expressed alternative splicing (DAS) events at the different development stages were identified using Log2FoldChage > 2 and *q*-value < 0.05. There were 10,807 differentially alternatively spliced genes and 14,912 genes belonging to differentially alternatively spliced events in the first stage compared with the second stage and 125 differentially alternatively spliced genes and 675 differentially spliced genes belonging to differentially alternatively spliced events in the second stage compared with the third stage. There were a total of 69 genes that were both differentially alternative splicing genes and that had differential alternative splicing events occurring in all three stages (Figure 2A). The corresponding gene families were matched according to the existing gene annotation files to obtain the gene families to which the differentially expressed genes belonged. It finds that the transcription factors (TF) were mainly concentrated in the bHLH family, AP2-EREBP family, MYB family, and MADS family (Figure 2B; Appendix A). By comparing the differentially expressed genes with the existing splicing factor family data of *Arabidopsis thaliana*, 17 splicing factor (SF) families were obtained, in which glycine-rich protein, 17S U2 snRNP, related to spliceosome, 35S U5-associated proteins, and SR protein played a crucial part (Figure 2B; Appendix A).

### 2.3. Gene Ontology and Kyoto Encyclopedia of Genes and Genomes Enrichment Analysis

In order to clarify the functional categories of differentially expressed genes during the development of *O. fragrans* fruit, we performed GO enrichment analysis on the differentially expressed gene sets of comparison between two adjacent periods (S1–S2 and S2–S3), and the top 10 GO items significantly enriched were listed. Compared with the first and second periods, DEGs mainly included lipid metabolic process, small molecule metabolic process, chloroplast, plastid, phosphotransferase activity, alcohol group as acceptor, kinase activity, and oxidoreductase activity (Figure 3A), and DAS events enrichment in small molecule metabolic process, chloroplast, plastid, phosphotransferase activity, alcohol group as acceptor, kinase activity, and oxidoreductase activity (Figure 3A). Following consideration of the first and second segments, both differentially alternatively spliced genes and differentially alternatively spliced events were enriched in small molecule metabolic process, chloroplast, plastid, phosphotransferase activity, alcohol group as acceptor, kinase activity, and oxidoreductase activity (Figure 3A). The third period was contrasted with the second period, with a significant concentration of DEGs in the mRNA metabolic process, alpha-amino acid metabolic process, mRNA process cell periphery and oxidoreductase activity (Figure 3B), and DAS events concentration on mRNA metabolic process, alpha-amino acid metabolic process, mRNA process, cell periphery, and oxidoreductase activity (Figure 3B). DEGs and DAS between the second and third stages were accompanied by the mRNA metabolic process, alpha-amino acid metabolic process, mRNA process, cell periphery, and oxidoreductase activity (Figure 3B). Distinctions between these three periods collectively enrich the oxidoreductase activity.

KEGG enrichment analysis of genes can predict the key metabolic pathways involved in genes. In order to understand the key metabolic pathways related to the development of *O. fragrans* fruit, we, respectively, carried out KEGG pathway enrichment analysis on the differential gene sets of three groups of controls. Contrasted to stage one (S1) and stage two (S2), DEGs augmented in porphyrin and chlorophyll metabolism, photosynthesis-antenna proteins, photosynthesis, carotenoid biosynthesis, and alpha-linoleic acid metabolism (Figure 4A). DAS events were enriched in MAPK signaling pathway-plant, peroxisome, and cyanoamino acid metabolism pathways (Figure 4A). Observing the second (S2) and third periods (S3), DEGs were enriched in porphyrin and chlorophyll metabolism, photosynthesis-antenna proteins, alpha-linoleic acid metabolism, and carotenoid biosynthesis (Figure 4B); DAS events were enlisted in alpha-linolenic acid metabolism, phagosome, and carotenoid biosynthesis (Figure 4B). KEGG analysis of the DEG and DAS genes exposed that most of the genes were enriched in porphyrin and chlorophyll metabolism, alpha-linoleic acid metabolism, and carotenoid biosynthesis pathways.

### 2.4. RT-PCR Validation of Differentially Expressed Alternative Splicing Genes

In order to verify the AS events detected by RNA-seq, we extracted RNA samples from *O. fragrans* fruit at three different stages and designed primers for four genes that produce AS events for RT-PCR analysis (Appendix A). For example, ofr.gene4802 has two corresponding transcripts, in which the expression level of the transcript gene4802-mRNA-1 is increasing in three periods, which is consistent with the transcript data we have measured (Figure 5C). Comparing the expression levels of the two transcripts shows that the transcript ofr.gene4802-mRNA-1 exerts a dominating feature in the development of *O. fragrans* fruit (Figure 5C). The expression level of the two corresponding transcripts of ofr.gene55649 was increased in all three periods (Figure 5D), and the expression level obtained from the experimental results is highly consistent with the data we measured. In comparison, the transcript BGI_novel_T008614 exerts a dominant role (Figure 5D).

## 3. Discussion

### 3.1. Up-Regulation of Alpha-Linolenic Acid Metabolism in O. fragrans Fruit Development

Alpha-linolenic acid content is continuously accumulated in plants, which is the substrate of fatty acid oxidation. It participates in the synthesis of volatile compounds and is an indispensable substance in the human body but can only be obtained from plants [17,18]. The more mature the fruit, the higher the content of alpha-linolenic acid, which has also been verified in tomatoes and *Plukenetia volubilis* [18,19]. The alpha-linolenic acid pathway was significantly enriched in our enrichment outcome (Figure 4). A total of 37 genes were enriched in the alpha-linolenic acid synthesis pathway in the transcriptome data of *O. fragrans* fruit development, which is composed of seven gene families (Appendix A). Lipoxygenase (LOX), hydroperoxide dehydratase (AOS), 12-oxophytodienoic acid reductase (OPR), and other gene families have a trend of increasing with fruit development.

Lipoxygenase (LOX) gene is the key enzyme gene in the first step of the transformation of α-linolenic acid into other substances [20]. In *O. fragrans* fruit, the LOX gene family showed an overall up-regulation trend, consisting of 11 genes (29.73%), among which the transcript BGI_novel_T021035 (Table 1) corresponding to ofr.gene 24,865 genes could be up to 10 times higher in the third-period stage (S3) compared to the first-period stage (S1) (Figure 5C). Addition of LOX expression can accelerate the softening process of fruit after ripening [21]. During the ripening process of *O. fragrans* fruit, the fruit pericarp becomes soft continuously (Figure 1A). Therefore, we hypothesized that the expression of the LOX gene was significantly up-regulated, which may result in the fruit pericarp keeping softening. The overall gene expression level of the alpha-linolenic acid metabolism pathway increased with fruit ripening (Figure 5C), which was consistent with the change in the alpha-linolenic acid metabolism pathway in peaches [22]. The increase in LOX gene expression promoted the increase in downstream reaction substrate content, and alpha-linolenic acid accumulated continuously in *O. fragrans* fruit, which accelerated the softening of the fruit pericarp.

### 3.2. Carotenoid Metabolism Affects the Color Formation of O. fragrans Fruit

The carotenoid metabolism pathway is an important pathway that affects fruit color. During the ripening of *Mangifera indica* and *Vitis vinifera*, the carotenoid content gradually decreases and the change in carotenoid content is closely related to the change in fruit color [23,24]. The overall carotenoid content was decreased continuously from 0.032 mg/g in the first period to 0.021 mg/g in the third period, which was consistent with the overall carotenoid expression changes in transcription data (Figure 6B), where a total of 78 genes, including 12 gene families, were enriched in the carotenoid pathway (Appendix A). The differential regulation of gene expression levels was more obvious in the middle and downstream genes of the carotenoid metabolism pathway. In *Mangifera indica*, the difference in α-carotene and β-carotene content was responsible for the different colors of the flesh [25]. Our study found that the gene expression level difference is more obvious after the production of α-carotene and β-carotene in the carotenoid biosynthesis pathway of fruit at different stages (Figure 6B); the expression of genes of α-carotene and β-carotene are highest in the first period of fruit and then gradually decrease, which may be the reason for the gradual purple color of the fruit.

The gene expressions of zeta-carotene isomerase (Z-ISO), lycopene epsilon-cyclase (LCYE), lycopene beta-cyclase (LCYB), zeaxanthin epoxidase (ZEP), and 9-cis-epoxycarotenoid dioxygenase (NCED) in the carotenoid metabolism pathway also showed a downward trend, in which low expression impedes carotenoid synthesis (Figure 6A, Table 2). As a key enzyme, Z-ISO determines the conversion of carotenoid synthesis, thus regulating fruit coloring [26]. The transcriptional regulation of Z-ISO expression has changed during tomato evolution, which may result in differences in fruit color [27,28]. In our study, the transcript expression level of Z-ISO in the second phase decreased by 1/7 compared with that in the first phase, and the expression level in the third phase was roughly the same as that in the second phase (Figure 6B). LCYE and LCYB may be the limiting step in carotenoid accumulation. In *Citrus sinensis*, the low carotenoid content was consistent with the low expression of LCYE in the green stage. LCYB controls the accumulation of carotenoids in the middle and lower reaches of *Citrus reticulata* [29,30] and has also been verified in *Cucurbita moschata*, *M. indica*, and *V. vinifera* [28,31,32]. In our discussion, the expression levels of LCYE and LCYB were consistent, the expression levels decreased to 1/6 or 1/10 from the first stage to the second stage, and the expression levels of the second and third stages were consistent, with lower transcription levels. The carotenoid content decreased continuously from the first stage to the third stage (Figure 6B), and the low carotenoid content coincided with the low expression of LCYE and LCYB; that low expression of LCYE and LCYB inhibited carotenoid accumulation in *O. fragrans*. ZEP is a key action point in the carotenoid pathway and an increase in ZEP gene expression during fruit ripening inhibits carotenoid accumulation, which has been demonstrated in *Vaccinium myrtillus* and *Prunus armeniaca* [33,34,35,36,37]. In *O. fragrans*, the expression level of ZEP can be reduced to 1/10, and the reduction is most obvious in the second stage of *O. fragrans* fruit development, which indicates that reduced carotenoid content due to low expression of Z-ISO, LCYE, LCYB, and ZEP genes may be an essential reason for the *O. fragrans* color transitions.

## 4. Materials and Methods

### 4.1. Sample Selection and Preparation

The fruit of the *O. fragrans* “Zi Yingui” was collected about 150 days after flowering in spring, and the samples were collected according to the color of the fruit skin, with the first period being full green, the second period being half green and half purple, and the third period being full purple. Three *O. fragrans* “Zi Yingui” trees of equal lengths and healthy growth were selected for collection on the campus of Nanjing Forestry University in close proximity to each other, and 10 g were taken from each of the three trees in each period as three replicates. After picking, we peeled off the skin and put it into liquid nitrogen to keep it fresh and took pictures of the samples under the body view mirror.

The determination method of carotenoid content in the pericarp of “Ziyingui” was to take 0.5 g of pericarp powder ground with liquid nitrogen into a 10 mL centrifuge tube, add 6 mL of 95% ethanol solution precooled at 4 °C, and extract it in darkness at 4 °C for 24 h, shaking many times during the period. The centrifuge was set at 4 degrees, 5000 rpm, centrifuged for 5 min, and the absorbance values of 1 mL of supernatant solution were measured at the wavelengths of 665, 649, and 470 nm, respectively. The formula for calculating carotenoid content is as follows [38]:Carotenoid content (mg/g) = ((1000 − OD_470_ − 2.05 × Chlorophyl A − 114.4 × Chlorophyl B)/245) × Total extraction solution (L)/Fresh weight of material (g)
Chlorophyll A (mg/g) = (13.95 × OD_665_ − 6.88 × OD_649_) × Total extraction solution (L)/Fresh weight of material (g)
Chlorophyll B (mg/g) = (24.96 × OD_649_ − 7.32 × OD_665_) × Total extraction solution (L)/Fresh weight of material (g)

### 4.2. Transcriptome Sequencing and Differential Alternative Splicing Analysis

Total RNA was treated by mRNA enrichment method; that obtained RNA was fragmented by interruption buffer, reverse transcription was conducted by random N6 primer, and cDNA double strand was synthesized to form double-stranded DNA. The resulting double-stranded DNA was blunt-ended and phosphorylated at the 5′ end, forming a sticky end with an “A” protruding from the 3′ end, and lighted with a bubble-like linker with a “T” protruding from the 3′ end. The ligation product was amplified by PCR with specific primers. The PCR product was thermally denatured into a single strand, and then that single-strand DNA was circularized by a section of bridge prim to obtain a single-strand circular DNA library, and the single-strand circular DNA library was sequenced on DNBSEQ.

Clean reads were obtained by filtering out reads with low quality, adapter contamination, and high content of unknown base N. Clean reads were then aligned to the reference genome for new transcript prediction, SNP, InDel, and differential splicing gene detection. The novel transcript with protein code potential was in addition to that reference gene sequence to form a complete reference sequence after the novel transcript was obtained, and the alternative splicing condition of the sample was detected by using rMATs [39]. DESeq2 R package was used to screen a differentially expressed gene and differentially expressed alternative splicing transcripts and to perform in-depth cluster analysis and functional enrichment analysis [40]. GO functional classification and KEGG enrichment analysis of alternatively spliced transcripts were performed using the pHYPER package [41] in R software. The *p*-value was FDR-corrected and, in general, a function with a Qvalue ≤ 0.05 is considered significantly enriched. The ORF of unigene was detected using getorf [42], then aligned to the transcription factor protein domain with hmmsearch [43] (data from TF), and then the unigene was characterized according to the transcription factor family characteristics described by PlantTFDB [44]. According to the existing splicing factor (SF) family information of *A. thaliana*, we compared the homologous genes of *O. fragrans* to identify the splicing factor family of *O. fragrans*.

### 4.3. RT-PCR Validation of AS Events

The total RNA of fruits was extracted with RNAprep Pure Plant Plus Kit (DP441, TIANGEN, Beijing, China) according to the manufacturer’s instructions. Then, 5 μg RNA was reversed transcribed by Evo M-MLV RT Premix for RT-PCR (AG11728, Accurate Biology, Changsha, China) for cDNA synthesis. The reaction steps of RT-PCR were as follows: 94 °C for 30 s, followed by 29 cycles of 98 °C for 10 s and 55 °C for the 30 s and 72 °C for 1 min, and 72 °C for 5 min. Three replicates were created for each sample, with *OfACT* as the internal reference gene, primers were designed using the prime3 website (https://primer3.ut.ee/ (accessed on 13 April 2023)), and all primers used in this experiment were listed in Appendix A. RT-PCR products were identified by 1.5% agarose gel electrophoresis.

## 5. Conclusions

The fruit color of *O. fragrans* gradually changed from green to purple during the development process and the fruit texture gradually softened. Based on this, transcription and alternative splicing analysis were conducted for the three developmental stages of *O. fragrans*. According to the type of alternative splicing events, the content of skipped exon type was the highest and most of the alternative splicing events occurred in the first two periods. Transcription factors were primarily concentrated in the bHLH family, AP2-EREBP family, MYB family, and MADS family. Splicing factors mainly center on glycine-rich proteins, 17S U2 snRNP, related to spliceosome, 35S U5 associated proteins, and SR proteins.

This study found that alpha-linolenic acid metabolism and carotenoid metabolism pathways may be important pathways during fruit development of *O. fragrans*, among which the up-regulation of LOX, AOS, and OPR may be the reason for the accumulation of alpha-linolenic acid. The down-regulation of Z-ISO, LCYE, LCYB, and ZEP may be the reason for the decrease in carotenoid content, resulting in the color of fruit changing from green to purple gradually. This study analyzed the fruit development process of *O. fragrans* from the perspective of alternative splicing, which provided new evidence for further study on the fruit development of *O. fragrans*.

## Figures and Tables

**Figure 1 ijms-24-08666-f001:**
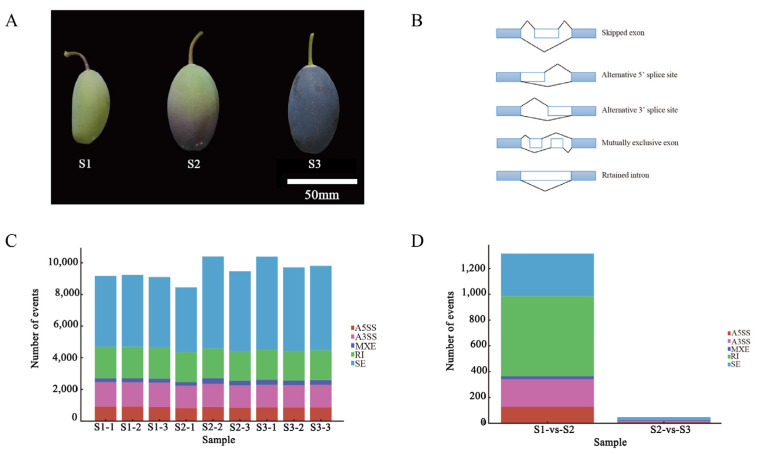
Phenotype and statistics of alternative splicing. (**A**) Phenotypes of *O. fragrans* fruits in three different periods; (**B**) schematic diagram of splicing events; (**C**) statistics of splicing events of different samples; (**D**) comparison of splicing events in different periods.

**Figure 2 ijms-24-08666-f002:**
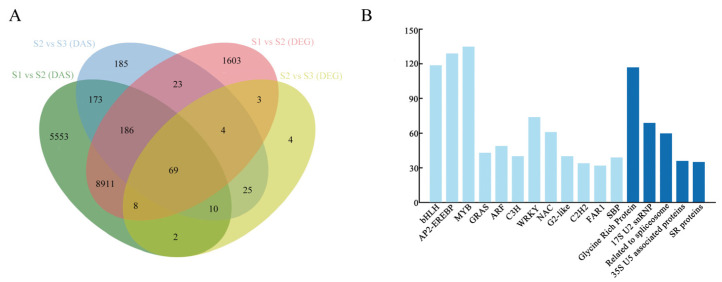
Differential alternative splicing analysis. (**A**) Venn diagram of differentially expressed genes (DEGs) and differentially expressed alternative splicing (DAS) events; (**B**) statistics of transcription factors and splicing factors of differential alternative splicing genes (number ≥ 50).

**Figure 3 ijms-24-08666-f003:**
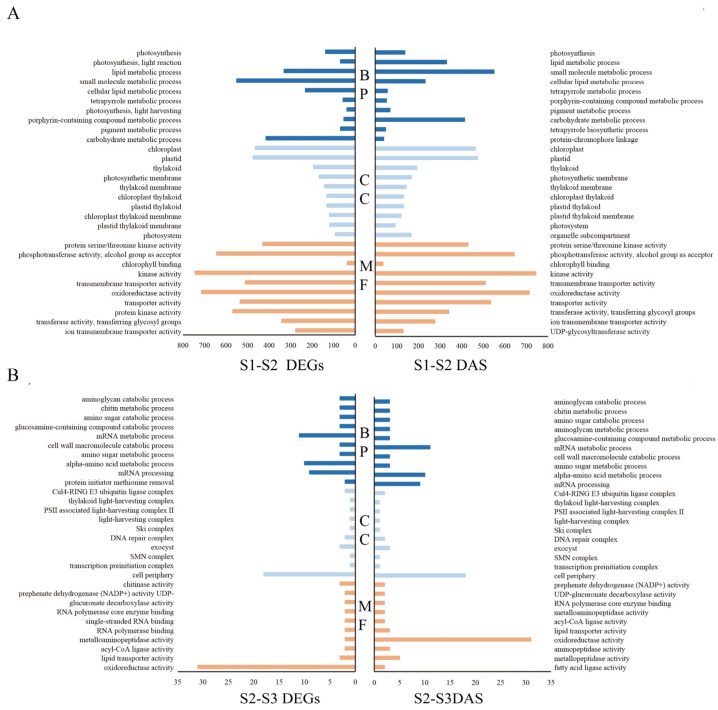
GO analysis of differentially expressed genes (DEGs) and differential alternative splicing (DAS) events. Dark blue, light blue, and orange represent the first 10 (**A**) GO analysis of differentially expressed genes in period one and period two; (**B**) GO analyses of differentially expressed genes in period two and period three. Biological processes (BP), cellular components (CC), and molecular functions (MF) down-regulated and up-regulated proteins in order by *p*-value.

**Figure 4 ijms-24-08666-f004:**
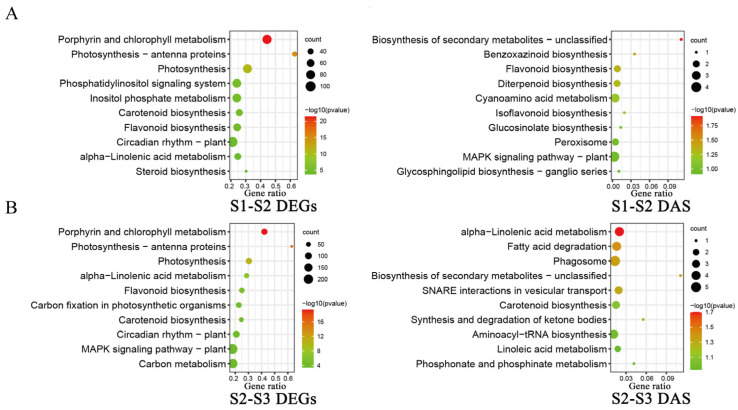
KEGG analysis of differentially expressed genes (DEGs) and differential alternative splicing (DAS) events. (**A**) Variations in KEGG analysis of differentially expressed genes and differential alternative splicing events in the first and second periods; (**B**) contrast of KEGG analysis of differentially expressed genes and differential alternative splicing events in the second and third periods.

**Figure 5 ijms-24-08666-f005:**
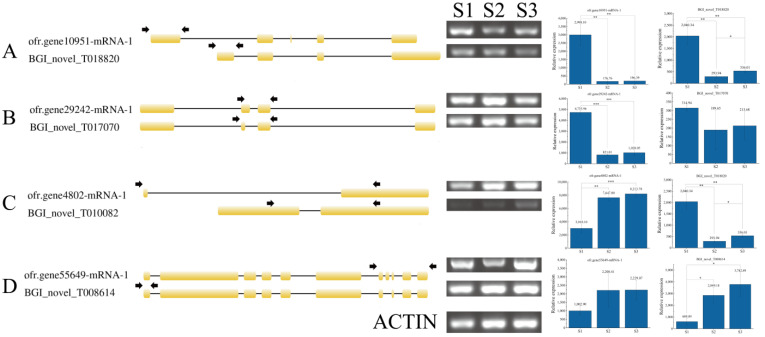
RT−PCR validation of differentially expressed genes, (**A**–**D**) represents differentially expressed genes. Genetic map, experimental validation, and transcriptome determination of the expression levels of the different transcripts over the three time periods; arrows represent primers designed for each isoform. Data are the mean ± stand error from three independent experiments. Significant differences are indicated with asterisks above the bars (* *p* ≤ 0.05, ** *p* ≤ 0.01 and *** *p* ≤ 0.001).

**Figure 6 ijms-24-08666-f006:**
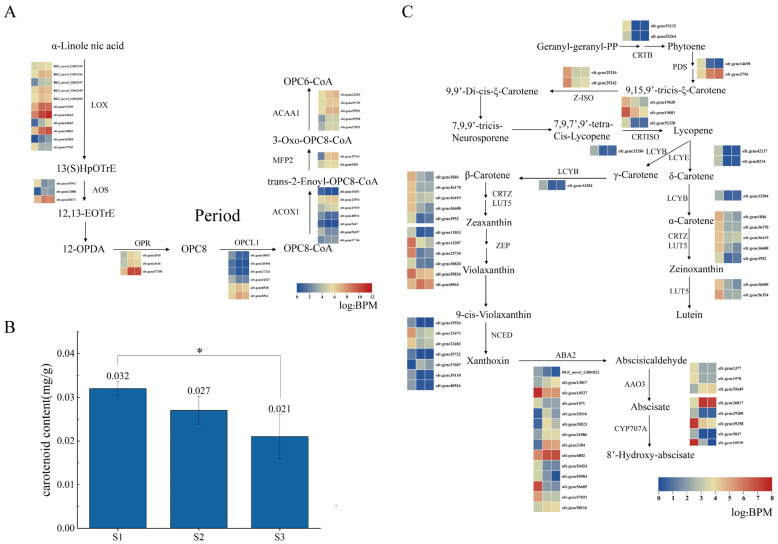
Pathway analysis. (**A**) Alpha-linolenic acid metabolism pathway and gene expression heatmap; (**B**) determination of carotenoid content; (**C**) carotenoid pathway and gene expression heatmap. Data are the mean ± stand error from three independent experiments. Significant differences are indicated with asterisks above the bars (* *p* ≤ 0.05).

**Table 1 ijms-24-08666-t001:** DEGs and DAS in the alpha-linolenic acid metabolism pathway.

Isoform	Gene	Gene-Family	S1	S2	S3
BGI_novel_T016400	BGI_novel_G002195	LOX	6701.00	12,170.67	10,720.67
BGI_novel_T016401	BGI_novel_G002196	LOX	9670.33	27,281.33	22,006.67
BGI_novel_T016402	BGI_novel_G002197	LOX	845.00	3869.33	4713.33
BGI_novel_T016404	BGI_novel_G002199	LOX	13,446.33	27,287.33	22,494.00
BGI_novel_T016405	BGI_novel_G002200	LOX	10,720.33	22,245.33	18,129.33
BGI_novel_T009589	ofr.gene19109	LOX	35,641.67	145,639.67	129,475.33
ofr.gene19109-mRNA-1	ofr.gene19109	LOX	17.33	8.33	53.33
BGI_novel_T021032	ofr.gene24663	LOX	79,012.67	330,175.00	522,090.33
ofr.gene24663-mRNA-1	ofr.gene24663	LOX	1.00	25.33	19.00
ofr.gene24665-mRNA-1	ofr.gene24665	LOX	20.67	228.33	691.33
BGI_novel_T021033	ofr.gene24865	LOX	320.67	1232.33	1758.67
BGI_novel_T021034	ofr.gene24865	LOX	801.00	6089.00	9685.67
BGI_novel_T021035	ofr.gene24865	LOX	6975.33	39,254.00	72,009.33
BGI_novel_T021036	ofr.gene24865	LOX	55.67	301.33	415.33
ofr.gene24865-mRNA-1	ofr.gene24865	LOX	79.33	286.33	3385.00
BGI_novel_T023460	ofr.gene34384	LOX	16.33	647.33	1578.33
ofr.gene34384-mRNA-1	ofr.gene34384	LOX	1.00	1.00	1.00
ofr.gene47520-mRNA-1	ofr.gene47520	LOX	12,844.67	2456.33	2501.00
BGI_novel_T018820	ofr.gene10951	AOS	2040.33	294.00	536.00
ofr.gene10951-mRNA-1	ofr.gene10951	AOS	2998.33	176.67	196.33
ofr.gene12088-mRNA-1	ofr.gene12088	AOS	70.67	1040.33	759.33
ofr.gene53671-mRNA-1	ofr.gene53671	AOS	14,221.00	46,525.67	40,584.67
ofr.gene3545-mRNA-1	ofr.gene3545	OPR	1682.00	6327.67	4652.67
ofr.gene3636-mRNA-1	ofr.gene3636	OPR	1753.00	7861.33	6424.67
BGI_novel_T003932	ofr.gene57358	OPR	13,911.00	163,878.67	141,732.00
ofr.gene57358-mRNA-1	ofr.gene57358	OPR	226.00	567.67	531.67

**Table 2 ijms-24-08666-t002:** DEGs and DAS in carotenoid metabolism pathway.

Isoform	Gene	Gene-Family	S1	S2	S3
BGI_novel_T017069	ofr.gene29216	Z-ISO	6647.67	1403.00	1603.67
BGI_novel_T017070	ofr.gene29242	Z-ISO	314.67	189.67	213.67
ofr.gene29242-mRNA-1	ofr.gene29242	Z-ISO	4735.67	820.67	1020.33
ofr.gene42137-mRNA-1	ofr.gene42137	LCYE	910.00	19.33	33.33
ofr.gene8214-mRNA-1	ofr.gene8214	LCYE	910.00	19.33	33.33
BGI_novel_T016725	ofr.gene33284	LCYB	1004.33	55.67	108.67
ofr.gene33284-mRNA-1	ofr.gene33284	LCYB	14.67	15.33	11.67
BGI_novel_T018783	ofr.gene11033	ZEP	1215.33	29.00	84.33
BGI_novel_T018237	ofr.gene12387	ZEP	7590.00	1044.67	629.33
BGI_novel_T001884	ofr.gene23734	ZEP	9843.33	54.00	370.33
BGI_novel_T001885	ofr.gene23734	ZEP	579.00	197.67	29.33
BGI_novel_T017743	ofr.gene30820	ZEP	1109.33	167.67	144.67
BGI_novel_T008623	ofr.gene55834	ZEP	3569.00	1294.00	1092.00
ofr.gene55834-mRNA-1	ofr.gene55834	ZEP	1727.33	533.33	610.00
ofr.gene8564-mRNA-1	ofr.gene8564	ZEP	2680.33	7073.67	4566.00

## Data Availability

The raw data of sRNA sequencing in this study are available in the National Center for Biotechnology Information (NCBI) Sequence Read Archive (SRA) database under BioProject PRJNA898340 (accession number SRP408527).

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
