# Peer review of "Alternative Splicing Analysis Revealed the Role of Alpha-Linolenic Acid and Carotenoids in Fruit Development of Osmanthus fragrans"

_ijms, 2023, doi:10.3390/ijms24108666_

Round 1

Reviewer 1 Report

Dear Editor/Authors

I revised the manuscript (ijms-2266575) titled: “Alternative Splicing Analysis Revealed the Role of alpha-Linolenic Acid and Carotenoids in Fruit Development of Osmanthus fragrans.

The paper is so difficult to understand, authors should present a paper easy for readers. The common name of this fruit species should be provided. The objectives of the work should be clear. In the material and methods section the choice of the stage of sampling is not justified. The Number of fruits, orgin, number repetitions… should be indicated. The extraction method not explained. The protocol of carotenoids should be explained. The description of methods use in this study should be presented in the past form. All the tables and figures should be presented in the results section.

Author Response

Thank you for your comments, the common name of the fruit has been added to the abstract article: O. fragrans ‘Zi Yingui’ (L19 and L82), this study is based on the fruit peel of cinnamon, the color of the peel changes from green to purple, so the samples were collected according to the color change of O. fragrans fruit represent the three key stages of laurel fruit development(L81-L85).

Information on the number of fruits, their origin, and the number of repetitions has been added (L314-L317) and the extraction method was described in the Materials and Methods section (L328-L366). Carotenoid content, an important part of the discussion, was used to determine changes in carotenoid content over three periods (L294). The descriptions of the methods used in this study have been changed to the past tense and most of the graphs in this paper are in the results section, with some graphs listed in the discussion section in conjunction with the need for discussion.

Reviewer 2 Report

The study focused on identifying DEGs and alternative splicing forms during the process of fruit development in O. fragrans.  Chloroplast/plastid-related processes were found to be altered during fruit development. Transcriptomes of three stages were analyzed and S1 vs S2 and S2 vs S3 stages were compared. S1 vs S3 comparison was not required. The MS can be accepted for publication after some minor changes. 

1. What is x axis? Mention it in figure 4.

2. line 183   gene66649 

The gene number does not match the one mentioned in figure 5D.

3. Label the three lanes in the gel image in figure 5. Also, label the y-axis of the graphs in Figures 5 and 6B.

4. Check the MS for minor grammatical mistakes.

Author Response

Thank you for your comments, I will reply to your comments one by one.

  1. The X axis is the Gene ratio, which is the number of genes as a proportion of all genes, and is indicated in Figure 4.
  2. It should be ofr.gene55649, which has been modified (L191).
  3. Markings and modifications have been made in Figures 5 and 6B.
  4. Already Double check for grammar issues (L18, L21, L45-L47, L73, L80, L83-L85, L89, L96, L97, L103, L119, L123, L140, L150, L160, L170, L171, L185, L189, L190, L191, L214, L219, L221, L266, L282, L283, L329, L330, L333, L342, L345, L366, L370, L371, L373, L378).

Reviewer 3 Report

In the submitted manuscript, the authors described transcriptome data from three different stages of fruit development of O.  fragrans were used to analyze the types and quantities of alternative splicing. Additionally analyzis of GO and KEGG were performed on the differentially spliced genes and differentially spliced isoforms. Authors also discuss the role of alternative splicing (AS) during fruit development or color shift and provide the basis for further research of O. fragrans.

The authors conducted extensively in silico analyses of the studied genes with AS , combined with the experimental validation of gene expression in transcript level by RT-PCR. The strong points of the manuscript is a very interesting topic The description of identified mRNA and alternative spliced transcripts is very interesting especially with regard to fruit development. Unfortunately, there was no mention of how the sequencing was carried out. Were these methods categorised as NGS? If so the current guidelines for expressional analysis are that experiments using RNA - Seq data to describe changes should contain the biological replicates (unless otherwise justified) herein authors have no mention this. Each biological replicate should be represented in an independent library, or each with unique barcodes, if libraries are multiplexed for sequencing. In this case, authors did not specify if the libraries were prepared independently.

Authors performed validation qPCR. Actually, the information given by the author in the manuscript about the replicas is not clear. According to the standard protocol, it  must contain three biological and three technical replicates. Herein there is lack of this information and thus it seems the qPCR analysis are not properly replicated.

The authors also did not indicate the accession number of the reference genome.

What about the control rerefernce gene in the qPCR?

Actually there is RT-PCR or qRT-PCR? What about the primers for qPCR techniques? Authors did not mention about software for primers design.

The description of qPCR quantification method is poorly described. It seems like the statistical methods are not well developed.

Author Response

Thank you. The sequencing methods for this experiment were summarised in the Materials and Methods section (L329-L355). Three replicates were set up for each stage and have been added to Materials and Methods, each biological replicate is a separate library and a separate library of files for each biological replicate is available in the data uploaded to NCBI.

The expression in Figure 5 of this paper is based on the results obtained from genotype-to-expression pairs measured in the transcriptome and compared with the results of RT-PCR (Real-time PCR), no qRT-PCR (quantitative real-time PCR) experiments were performed. Three replicates were created for each sample, with OfACT as the internal reference gene, primers were designed using the prime3 website (https://primer3.ut.ee/), and the designed primers are given in Table S5.

Reviewer 4 Report

The bibliography must be put in alphabetical order and The Material and method section should be before the Results section.

On the other hand, the research topic  is very interesting and supports the specialists in order to explain some processes related to fruit ripening. It is a deep research and should be continued for other species of interest for consumption.

Author Response

Thank you for your approval of our articles. Firstly, the format of the article is written strictly in accordance with the format of the journal. In the standard format provided, references are filled in according to the sequence order that appeared in the article and the material and methods should be placed at the end (https://www.mdpi.com/journal/ijms/instructions). The group is focused on the study of Osmanthus fragrans, with subsequent studies on the fruits of other species as well.

Round 2

Reviewer 1 Report

I see that authors considered all the needed comments. Responses were included in the text.

Author Response

Thank you for your approval of our revisions.

Reviewer 2 Report

The MS can be accepted for publication.

Author Response

(The authors gave the same response as above.)

Reviewer 3 Report

The authors responded to my comments and amendments were introduced. Nevertheless, I have some questions, and the answers to these questions should be included as information in the presemted manuscript:

Can you specify how many libraries in total you have sequenced?

What was the quality of the sequencing?

Can you specify what genome was used to map the reads?

Can you cite what was the quality of the mapping of the reads?

Has the genome been previously annotated?

How were transcript functions determined whether from genome annotation or by another method?

Author Response

Thank you very much for your careful and conscientious advice on our manuscript, we appreciate your time and effort once again; thank you very much for your question and apologise that our first response did not fully answer your query, here is our specific response to this question:

1.Can you specify how many libraries in total you have sequenced?  

A total of nine databases were sequenced, with three replicates for each sample, which can be found in the NCBI upload (L85-L87: In this project, a total of 9 samples were tested using publicthe DNBSEQ platform, and the samples at each stage had three biological replicates, with an average output of 6.41 Gb of data per sample.) (L407-L409: The raw data of sRNA sequencing in this study are available in the National Center for Biotech-nology Information (NCBI) Sequence Read Archive (SRA) database under BioProject PRJNA898340 (accession number SRP408527). )

2.What was the quality of the sequencing?

The sequencing quality is good and has been included in the article (L87-L90: A total of 417,176,396 sequencing tags were obtained, and an average of 42,706,801 tags were obtained for each sample. After filtering, the tags with sequencing quality higher than Q20 of each sample were more than 96% (Table S1)), the specific sequencing quality results are listed below and have been tabulated (Table S1).

TableS1 Reads quality statistics after data filtering

Sample Name

Raw Tag Count

Clean Tag Count

Clean Reads Q20 (%)

Clean Reads Ratio (%)

Total Mapping (%)

S1_1

45573892

42085678

96.78

92.35

87.17

S1_2

47326734

43431238

96.88

91.77

86.51

S1_3

47326734

43191314

96.84

91.26

86.18

S2_1

47326734

43233868

97.06

91.35

86.65

S2_2

45573892

42042568

96.74

92.25

86.91

S2_3

45573892

42679382

96.63

93.65

87.55

S3_1

45573892

42072286

96.78

92.32

87.31

S3_2

47326734

43373124

96.85

91.65

87.58

S3_3

45573892

42251748

96.79

92.71

87.83

3.Can you specify what genome was used to map the reads?  

The genomic data of laurel sequenced by our group is not yet published, so we cannot give detailed examples here, so I hope you will understand.

4.Can you cite what was the quality of the mapping of the reads?

The quality of the mapping of the reads has been added to Table S1.

5.Has the genome been previously annotated?

The unpublished database files of our group have corresponding gene annotation files, as mentioned above, the genomic data of laurel sequenced by our group, but they are not published yet, so they are not specifically cited here, I hope you can understand.

6.How were transcript functions determined whether from genome annotation or by another method?

Due to the existence of variable splicing, the same mRNA precursor can generate multiple transcripts by different splicing methods. We match the transcripts to the corresponding genes and interpret the function of the transcripts based on the genome annotation file.

Reviewer 4 Report

I suggested that  the section Materials and Methods should follow   The Introduction and The bibliography must be put in alphabetical order, therefore I agree with minor revision.

Author Response

We strictly follow the writing order and requirements of your journal (IJMS | Instructions for Authors (mdpi.com))and the latest publications are arranged in such a way that the materials and methods precede the conclusions and the references are arranged in the order in which they appear(IJMS | Special Issue : Alternative Splicing: From Abiotic Stress Tolerance to Evolutionary Genomics (mdpi.com)), so we hope you will understand.

Round 3

Reviewer 3 Report

The authors answered my questions for the most part. On the other hand, unfortunately, there is no information about the reference genome. This is essential and at least the genomic sequence should be included in the database and have a reference number.

Author Response

The genomic data of Osmanthus fragrans sequenced by our group is not yet published, so we cannot give detailed examples here, but the reads quality statistics after data filtering has been added to Table S1 and complemented with transcript sequence in Table S6, so I hope you will understand.